# Determinants of lifestyle counseling and current practices: A cross-sectional study among Dutch general practitioners

**Lisanne Kiestra**[1]☯, **Iris A. C. de Vries**[2], **Bob C. Mulder**☯[1]☯ *

**1** Strategic Communication, Department of Social Sciences, Wageningen University & Research, Wageningen, The Netherlands, **2** Vereniging Arts en Leefstijl, Utrecht, The Netherlands

☯ These authors contributed equally to this work.
* bob.mulder@wur.nl

**Data Availability Statement:** All relevant data are within the paper and its Supporting Information files.

**Funding:** The author(s) received no specific funding for this work.

## Abstract

This study aimed to examine the amount of lifestyle counseling that Dutch general practitioners (GPs) generally provide to their patients, as well as the behavioral determinants of their lifestyle counseling practices. Lifestyle counseling was defined and operationalized through the 5As model (i.e. Assess, Advise, Agree, Assist and Arrange), while determinants were based on an adapted version of the theory of planned behavior. A cross-sectional study was conducted among a sample of 198 GPs, using an online survey questionnaire for collecting data. The results showed that 79.3% of the GPs assessed patients' current lifestyle often or always, while 60.1% reported they often or always assessed patients' motivation to improve their lifestyle. Depending on the lifestyle behavior, Advising to improve lifestyle ranged from 42.5% (sleep) to 92.4% (smoking), while Agree to set goals ranged from 21.7% (sleep) to 46.9% (smoking). Assisting patients to overcome barriers to lifestyle changes varied per patient barrier, ranging from lack of financial resources (25.7%) to stress (81.8%). The findings from the linear hierarchical regression revealed that GPs' self-efficacy ($\beta = .46$, $p < .001$), patient norm ($\beta = .21$, $p < .001$), and attitude ($\beta = .20$, $p < .05$) were the determinants with the strongest associations with lifestyle counseling. The full model explained 47% of the variance in counseling lifestyle. Implications for supporting GPs to counsel patients about their lifestyle are discussed.

## Introduction

Chronic diseases are estimated to account for 90% of all deaths in the Netherlands [1] with cancer, cardiovascular diseases, respiratory diseases, and Alzheimer's and other dementias as the leading causes of death [2]. Given the role of behavioral risk factors in developing these chronic diseases [3], lifestyle is recognized as an important factor that should be addressed in their prevention [4]. A healthy lifestyle is defined as: 'a way of living that lowers the risk of being seriously ill or dying early [5], usually referring to health behaviors that include nutrition, physical activity, sleep, stress, tobacco use and alcohol use [6].

**Competing interests:** The authors have declared that no competing interests exist.

Lifestyle counseling by general practitioners (GPs) is important [7], and seems to be more effective than outsourcing counseling to a health coach or specialist, partly because patients view GPs as the most trusted source of health information [8]. Indeed, a British study [9] showed that GPs can already be effective with a 30 seconds behaviorally-informed, opportunistic intervention to reduce weight. However, studies show high variation in counseling practices between GPs. Typically, lifestyle behaviors are discussed in only a minority of the consultations, as it is not part of standard procedures. When GPs do counsel lifestyle, variation is also apparent in the type of health behaviors that are discussed, and in the lifestyle counseling elements or methods used by the GP [10].

According to the Dutch College of General Practitioners (NHG), GPs have an important signaling and advisory role to stimulate patient lifestyle behaviors [11]. However, due to lack of time and resources such as perceived counseling efficacy, there is large variation among GPs as to how and how often these guidelines are applied [10]. As a result, the variation between GPs in their counseling practices can be explained by underlying behavioral determinants of lifestyle counseling, such as their attitude towards lifestyle counseling, and perceived barriers and facilitators of counseling lifestyle [12,13]. For example, lack of time and reimbursement, and (perceived) skills to discuss lifestyle, as well as low patient motivation are often mentioned by GPs as barriers [14,15].

To date, only small-scale qualitative research has examined GPs lifestyle counseling practices and the underlying determinants [10,14]. Therefore, the aim of the present research is to quantitatively examine current GP lifestyle counseling practices, as well as the determinants thereof. This includes examining how often GPs provide lifestyle counseling during standard consultations; which particular lifestyle behaviors are discussed; and the theoretical elements of lifestyle counseling that are then applied; as well as GPs' attitudes, and other behavioral determinants that may explain variability in lifestyle counseling practices. Results from this study may serve multiple goals; first and foremost, knowledge about current lifestyle counseling practices of GPs may inform policy decisions about using and stimulating these practices in preventive or curative ways. Second, the analysis of determinants can be used for the development of interventions that support and increase GPs' counseling practices.

## Materials and methods

### Study design and participants

Cross-sectional data were collected by inviting GPs to participate in a self-administered survey questionnaire study. This research was conducted in collaboration with 'Vereniging Arts and Leefstijl' ('Cooperation Physician and Lifestyle'). This is an independent association that is committed to supporting healthcare professionals in applying lifestyle medicine. In March 2019, the survey invitation was sent to different GP groups; either directly via GP mailing lists, or by publishing the survey invitation in a GP or care group's monthly newsletter or practice update. The invitation to participants contained a hyperlink to the online survey questionnaire. Members of 'Vereniging Arts en Leefstijl' were directly accessible via email, as was a separate and national GP sample from research institute NIVEL. After two weeks, a reminder was sent to these two samples; sending a reminder to the other samples was unfortunately not feasible.

In total, approximately 1660 GPs were approached, and after four and half weeks, 265 respondents had opened the questionnaire. To increase the completion rate, the introduction of the questionnaire stated that every 25[th] respondent who completed the questionnaire would receive a gift voucher of 25 euro. In total, 200 GPs completed the questionnaire (response rate: 12%), 198 of whom were included in the analysis. Two GPs were excluded because they worked less than two dayparts per week.

## Data collection

Before data collection, a pilot study was held with five GPs to test the draft survey questionnaire. These GPs were involved in 'Vereniging Arts en Leefstijl' and had knowledge of lifestyle medicine. They. They received an email which contained a hyperlink to the online survey questionnaire and completed the questionnaire without the presence of the researcher. The aim was to collect critical comments in order to improve the questionnaire, which they send back by email. The researcher contacted them by telephone to clarify their feedback if necessary. After processing their feedback, for which they were rewarded with a gift voucher of 50 euro, the questionnaire was finalized and published using Qualtrics online survey tool.

The questionnaire consisted of three sections: the first section contained measures of lifestyle counseling practices, the second section contained measures of the determinants of those practices, and the third section contained questions pertaining to demographic and background variables (S1 and S2 Files).

## Measures

**Lifestyle counseling practices.**   Lifestyle counseling practices were defined and measured using the 5As framework [16]. The 5As framework serves to describe, teach and evaluate lifestyle counseling in primary care settings [17–19]. It is a series of interrelated and iterative counseling steps, called Assess, Advise, Agree, Assist, Arrange. Each step gives direction to the development of a personal action plan for the patient and the steps together predict the effectiveness of counseling [19,20]. In that sense, the 5As model presents the 'ideal' approach to counseling patients towards a healthy lifestyle.

Each of the 5As was measured separately, with items either directly derived from, or based on, validated scales [19–23]. All items employed a five-point Likert scale ranging from 'never' to 'always' as the response format. Based on Cronbach's alpha, reliability across all 22 items for the full 5As scale was good with $\alpha = .77$.

Assess was measured with two items: 'How often do you ask your patients about their lifestyle?' and 'How often do you ask whether patients are motivated to change their lifestyle?'. Advise was measured with one main question, but with six items pertaining to six lifestyle behaviors: 'Based on the assessment, how often do you advise your patients on the following lifestyle habits?'. The six lifestyle habits included smoking, alcohol use, nutrition, physical activity, sleep, and stress. The same lifestyle habits were measured with six items for the third A; Agree to collaboratively set goals: 'When you advise your patients, how often do you set concrete goals together to change the following lifestyle habits?'. Assist was measured with one main question pertaining to seven potential barriers, each with a separate item: 'How often do you discuss the following factors that may (possibly) present a barrier to a healthy lifestyle for patients?'. The seven barriers were stress, temptations, lack of time, lack of knowledge, lack of motivation, lack of financial resources, and lack of confidence. Finally, Arrange was measured with one item: 'When you give your patients lifestyle advice, how often do you provide follow-up support to them, for example, follow-up appointment, follow-up call, and/or medication reduction?'.

**Determinants of counseling practices.**   The determinants of lifestyle counseling practices, as well as their measurement, were derived from prior studies in this domain [13,14,24], and the theory of planned behavior [25]. The theory of planned behavior has been widely used to predict health behaviors of diverse patient populations [26–29], and–to a much lesser extent–to predict provider counseling behaviors [12]. According to the theory of planned behavior, the main determinants of behavioral intention are attitude, subjective norms, and perceived behavioral control. For each determinant, specific underlying and beliefs were measured. Regarding attitude, cognitive and affective beliefs were measured with seven items on a

semantic-differential scale (Cronbach's α = .89). All items started with 'I think discussing life-style is. . .', followed by, for example, the antonym pair 'easy-difficult' that formed the extremes of a seven-point scale.

Three separate social norms were measured, each with a single item that was rated on a five-point Likert scale ranging from 'strongly disagree' to 'strongly agree'. These were the per-sonal norm that GPs should discuss lifestyle; the perceived descriptive norm that other GPs discuss lifestyle; and the perceived patient norm that patients expect the GP to discuss lifestyle ("I believe patients expect me to discuss lifestyle" [13].

The last category of determinants included self-efficacy, and self-reported lifestyle. Self-efficacy towards counseling lifestyle was measured with two items on a five-point Likert scale ranging from 'not at all' 'to a very high degree' (Cronbach's α = .85). An example item is: 'I can motivate patients to improving their lifestyle'. Self-reported lifestyle was rated on a scale from 1 to 10.

Finally, GPs were asked with a single open question to list factors that would motivate or help them to discuss or counsel lifestyle more often.

**Background variables.**    The last section of the questionnaire measured the following back-ground variables: age, gender, number of working days, postal code of practice, the type of practice, and whether they know 'Arts en Leefstijl' and/or are a member. Working days were measured because we expected that GPs who work less have lower self-efficacy, as they see their patients less often. Considering type of practice, it could be possible that GPs of one or more types of practices counsel lifestyle more often than GPs of other types of practices. The same holds for knowing or being member of 'Vereniging Arts en Leefstijl', as it is likely that members counsel lifestyle more often than the non-members.

**Statistical analysis.**    SPSS version 25 was used for statistical analysis. For all scales, mean item scores were averaged into a single score per determinant, after checking scale reliability by calculating Cronbach's alpha.

To examine the relationships between lifestyle counseling and its determinants, bivariate correlations were computed. Next, multivariate associations were tested by means of a hierar-chical regression analysis with counseling lifestyle as the dependent variable (DV). Associa-tions with potential background variables were examined, and when background variables were significant (p < .05) in the first model, they were maintained as independent variables in the first block of the hierarchical regression. Self-reported lifestyle was entered in the second block and the other behavioral determinants, which were attitude, subjective norms, and per-ceived behavioral control, were entered in the third block of independent variables.

**Ethics approval and consent to participate.**    Data for this study were collected according to guidelines of the Association of Universities The Netherlands [30]. Approval from a medi-cal-ethical committee was not required, as this is not required when conducting survey ques-tionnaire research in the Netherlands [31,32]. Confidentiality was assured by using self-administered and anonymous questionnaires. The introduction of the questionnaire stated that the research aim of the study, and announced filling in was voluntary and anonymous. By clicking on the answer 'Yes, I agree with that', participants gave their informed consent. When participants clicked on 'No, I do not agree with that', a message came up that announced they could close the questionnaire.

## Results

### GP characteristics

Of the 198 participants included in the analysis, 140 were female (71%). The mean age of the sample was 46.8 years (*SD* = 10.0, range: 28–71 years). The mean number of dayparts partici-pants worked was 7.0 (SD = 1.8), ranging from 3 to 12 dayparts.

The participants worked in different types of practices, but most of them worked in an independent duo practice (24.2%), followed by solo practice (22.2%), independent group practice (20.7%), solo practice in health center (13.6%), group practice in health center (11.1%), and duo practice in health center (8.1%). Of the 198 participants, 74 (37.4%) were member of 'Vereniging Arts en Leefstijl'.

## Lifestyle counseling practices

79.3% of the GPs assessed patients' current lifestyle, while 60.1% reported they assessed patients' motivation to improve their lifestyle (see Table 1 for all percentages that each of the 5As was applied 'often' or 'always'). Advise and Agree were scored for six lifestyle behaviors separately: GPs advised most often about smoking (92.4%), followed by physical activity (79.3%) and alcohol use (69.2%). Sleep was the least advised lifestyle behavior (42.5%). A similar pattern was found for agreeing on goals to improve lifestyle behaviors. GPs most often agreed on goals for (quitting) smoking, while they agreed the least on goals for sleeping.

Concerning Assist, stress was the most frequently discussed barrier to assist patients (81.8%), followed by lack of motivation (53.0%), lack of time (47.8%), temptations (41.0%), lack of (self) confidence, and lack of time (both 31.8%), and lack of financial resources (25.7%).

Finally, almost half of the GPs indicated to arrange a follow-up meeting, or refer to another health care (or lifestyle) professional to improve patients' lifestyle. With regard to referring, the dietician was the professional to which GPs refer to most often (67.2% 'often' or 'always'), followed by the practice nurse (62.7%), the physiotherapist (39.4%), and the psychologist (12.6%). The registered lifestyle coach (BLCN) (4.0%) and the medical specialist (0.5%) were the professionals to whom GPs referred the least often.

In total, 60% of the GPs would like to discuss or advise lifestyle more often. In reply to the open question, GPs mentioned several factors that would help or motivate them to discuss or advise lifestyle more often. Facilitating external factors were having more time, which was by far the most frequently mentioned facilitator. This was followed by a need for supporting tools, and information materials such as leaflets. Clear and practical guidelines was another frequently mentioned facilitator. At last, some GPs reported a need for scientific evidence for the effectiveness of lifestyle, financial compensation, appropriate referral options and collaboration with other disciplines. As one GP formulated this:

*"More time, teamwork within health care and stimulating patients' motivation by public campaigns"*

**Table 1. Counseling practices: Percentages of applying the 5As 'often' or 'always'.**

| 5As component | in general (single item) | nutrition | physical activity | smoking | sleep | alcohol use | stress |
|---|---|---|---|---|---|---|---|
| Assess lifestyle | 79.3 | | | | | | |
| Assess motivation | 60.1 | | | | | | |
| Advise | | 60.6 | 79.3 | 92.4 | 42.5 | 69.2 | 62.2 |
| Agree | | 32.8 | 39.4 | 46.9 | 21.7 | 33.8 | 30.3 |
| Assist | 25.7–81.8[a] | | | | | | |
| Arrange follow-up | 44 | | | | | | |
| Arrange referring | 45[b] | | | | | | |

[a] measured with 7 items; each corresponded with a barrier: stress, lack of finances, lack of time, lack of motivation, lack of confidence, lack of knowledge, temptations.
[b] only 'often', as 'always' was never scored for this item.

Secondly, GPs reported personal facilitating factors, specifically gaining more knowledge and motivation, e.g., through attending courses. Facilitating patient factors were not mentioned often. Only a few GPs mentioned that motivated patients would help or motivate them. This is another answer to the open question to illustrate:

> "Training, time, guidelines, standards and financial compensation for us. If we as GPs have to pick it up, it must be really well organized!"

### Determinants of lifestyle counseling

**Correlations.** Pearson correlations show that the strongest correlates of counseling lifestyle (i.e. applying the 5As) were self-efficacy ($r = .62$, $p < .01$) and attitude ($r = .50$, $p < .01$), and the weakest correlation was self-reported lifestyle ($r = .19$, $p < .01$) (see Table 2 for a complete overview of correlations). Of the three norms, the personal ($r = .36$, $p < .01$) and patient norm ($r = .35$, $p < .01$) were associated with lifestyle counseling, but not the descriptive norm. Also noteworthy is the positive association between self-reported lifestyle and self-efficacy ($r = .31$, $p < .01$), indicating that a more healthy lifestyle is associated with higher self-efficacy to provide lifestyle counseling.

**Multivariate associations.** The regression analysis that aimed to test associations with background variables showed that only being a member of 'Vereniging Arts en Leefstijl' and gender were significantly associated with lifestyle counseling. Therefore, these variables were maintained as covariates throughout the main regression analysis. Age and working days were not significantly associated with lifestyle counseling in any of the models and were therefore excluded.

In model 1, being a member of 'Vereniging Arts & Leestijl' and gender explained 9% of the variance in lifestyle counseling (see Table 3). When self-reported lifestyle was added, the explained variance increased to 12%. The full model, including attitude, subjective norms and self-efficacy explained 47% of the variance in lifestyle counseling. Self-efficacy ($\beta = .46$, $p < .01$) showed the strongest relationship with lifestyle counseling, followed by patient norm ($\beta = .21$, $p < .01$) and attitude ($\beta = .20$, p < .05). Personal norm and descriptive norm were not associated with lifestyle counseling.

Compared to model 2, membership and self-reported lifestyle were not associated with lifestyle counseling anymore. This suggests that these associations can be explained by attitude, perceived patient norm and self-efficacy.

### Discussion

The aim of the present study was to examine the extent to which GPs provide lifestyle counseling to their patients in general. Second, and for the first time, we examined the behavioral

**Table 2. Pearson correlations between lifestyle counseling and its determinants (N = 198).**

|  | 2. | 3. | 4. | 5. | 6. | 7. |
|---|---|---|---|---|---|---|
| 1. 5As | .50** | .36** | -.04 | .35** | .62** | .19** |
| 2. Attitude | - | .63** | -.10 | .31** | .51** | .07 |
| 3. Personal norm | | - | .12 | .41** | .38** | .15* |
| 4. Descriptive norm | | | - | .41** | .21** | -.08 |
| 5. Patient norm | | | | - | .20** | .06 |
| 6. Self-efficacy | | | | | - | .31** |
| 7. Self-reported lifestyle | | | | | | - |

**$p < 0.01$,
*$p < 0.05$.

**Table 3. Regression analysis to test associations between determinants and lifestyle counseling (DV).**

|  | *B* | *β* | *SE* | *t* | *R²* | *R² change* |
|---|---|---|---|---|---|---|
| DV: lifestyle counseling |  |  |  |  |  |  |
| Model 1: |  |  |  |  | .09 |  |
| Member | .25 | .26 | .07 | 3.67** |  |  |
| Gender | -.09 | -.09 | .07 | -1.28 |  |  |
| Model 2: |  |  |  |  | .12 | .04 |
| Member | .25 | .26 | .07 | 3.80** |  |  |
| Gender | -.09 | -.09 | .07 | -1.34 |  |  |
| Self-reported lifestyle | .10 | .20 | .03 | 2.93** |  |  |
| Model 3: |  |  |  |  | .47 | .34 |
| Member | .07 | .07 | .06 | 1.11 |  |  |
| Gender | -.04 | -.04 | .06 | -.67 |  |  |
| Self-reported lifestyle | .01 | .03 | .03 | .51 |  |  |
| Attitude | .10 | .20 | .04 | 2.60* |  |  |
| Personal norm | -.04 | -.07 | .05 | -.87 |  |  |
| Descriptive norm | .01 | .02 | .04 | .23 |  |  |
| Patient norm | .12 | .21 | .04 | 3.25** |  |  |
| Self-Efficacy | .31 | .46 | .05 | 6.85** |  |  |

**p < 0.01,

*p < 0.05.

determinants that explain variance in the amount of lifestyle counseling GPs generally provide. Lifestyle counseling was operationalized through the 5As model, which is a counseling model that comprises the steps to guide patients towards lifestyle behavior change. Results showed that almost 80% of the GPs assessed patients' lifestyle often or always during consultations. This indicates that in this sample, GPs are highly inclined to discuss lifestyle with their patients, which is in line with earlier research [13]. GPs even reported that they would be willing to provide more lifestyle counseling, if certain external (e.g., more time, clear guidelines and protocols) or personal facilitators (e.g., more knowledge) would be available; and these factors have been identified before [33].

Compared to Assess, GPs report less frequent application of giving Advice and, and even lower frequencies for goal setting (Agree). Helping patients to overcome barriers (Assist) showed large variation as to the type of barrier being discussed. This is in line with studies on the use of the 5A's, specifically for weight loss counseling, which show that physicians most frequently apply Assess and Advise, and rarely Agree, Assist or Arrange [19,34]. In the present study, however, 44% of GPs used Arrange 'often' or 'always'. This could be explained by the fact that Arrange a follow-up meeting is a standard procedure in the Netherlands.

Examining differences in lifestyle behaviors revealed that smoking was the most advised lifestyle habit, and also resulted in the most goal setting (Agree). This was followed by physical activity and alcohol use. In a comparable study [10], physical activity was the most discussed lifestyle habit and alcohol use the least discussed during primary care consultations. Also, stress and sleep were not included in their research, similar to many other studies in this domain [13,21,35,36]. This shows that more emphasis is put on the conventional lifestyle habits, thus less on stress and sleep, although it can be argued that sleep and stress should also be addressed [37].

With regard to the determinants, the full model explained 47% of the variance in lifestyle counseling, and since this is the first study to examine these determinants, no comparison data

is available. The theory of planned behavior is often used to explain health behavior and literature shows that they explain an averaged 34% of health behaviors [38], but of course this is not the same as counseling those behaviors.

In order of magnitude, self-efficacy, attitude and patient norm contributed significantly to explaining variability in GP's lifestyle counseling. In other words, GPs who feel confident that they can provide lifestyle counseling, and who believe that lifestyle counseling is useful and motivating to do, as well as believe patients expect them to discuss lifestyle are much more likely to counsel. The predicting role of self-efficacy in lifestyle counseling has been reported before [39], and is in line with social cognitive theory [40], which posits that self-efficacy is a pivotal determinant of behavior.

Noteworthy is that of the three norms, only the perceived norm that patients expect the GP to discuss lifestyle is significantly associated with lifestyle counseling. In contrast, the perceived personal norm that counseling is part of the GP role, and the descriptive norm that other GPs discuss lifestyle do not show a relationship with counseling. The important role for patient acceptance or even expectations towards lifestyle counseling has also been found in earlier studies [13,41].

In contrast to earlier studies [42–44], the present study did not find GPs' self-reported lifestyle to be an important independent associate of lifestyle counseling, as including the TPD determinants rendered self-reported lifestyle non-significant. It is therefore likely that healthier GPs have a more positive attitude and a higher self-efficacy towards lifestyle counseling

Findings of the present study indicate that lifestyle counseling could be stimulated by two distinct, but complementary, approaches. First, training programs for GPs should focus on increasing their attitude, perceived patient norm, and especially self-efficacy. Self-efficacy can be increased in several ways, one being mastery experiences that are created through guided practice with feedback [45], e.g., through role playing [46]. Both during medical school and/or postgraduate courses, such approaches could increase self-efficacy. Trainings could also motivate GPs and increase perceived outcomes of lifestyle counseling, thus bolstering positive attitudes. Finally, the norm that patients actually expect GPs to discuss lifestyle could be conveyed in such trainings.

Second, the facilitating factors could be improved, for example by through offering a structured management tool for obesity care in general practice (e.g., [47]). Other tools could include manuals, leaflets or apps for GPs or patients to further support counseling and subsequent behavioral change among patients. Since more time is the most frequently mentioned facilitator, a change in the reimbursement of hours of healthcare professionals is essential, whereby a shift takes place from the reimbursement of time for illness to the reimbursement of time for prevention and lifestyle.

The strengths of this study include the relatively large sample size, the explicit theory-base, and that it focused on the combination of the counseling practices of Dutch GPs and its determinants. As such, it was the first study that examined the influence of determinants on lifestyle counseling practices. In addition, in contrast to many other similar studies, various lifestyle habits were included to examine counseling, instead of focusing on only one lifestyle habit.

There are also several limitations to this study. First, because of the cross-sectional nature of this study, causality could not be determined. Consequently, conclusions about the causal pathway of determinants on counseling lifestyle cannot be drawn. For instance, it could be that providing lifestyle counseling increases attitude and self-efficacy, as well as the perception that patients expect it.

Second, self-reported data are prone to memory bias and/or social desirability bias. The latter means that the sample overestimates itself, and reports too positively about their counseling practices.

Third, using a convenience sample could have led to selection bias. It is likely this sample had an overrepresentation of GP's being engaged in lifestyle counseling. Female GPs were overrepresented as well. Possibly, this is related to the low response rate (12%, which is lower than the expected 25% [48]. Therefore, results should be interpreted cautiously; specifically the percentages of GPs who indicate using the 5As 'often' or 'always' can present an overestimation as compared to the general GP population. However, overestimating the use of the 5As is assumed to be equal across As, meaning that relative differences between the As are informative. Also, as a selection bias may result in a restriction of range, the reported associations between determinants and lifestyle counseling are more likely to be underestimated, thus are even stronger in reality.

Finally, the 5A's model also has some limitations. We used it to assess the quality of counseling on basis of the frequency with which the 5A's are applied. As a result, the way GPs apply the A's, especially their communication styles, could not be examined. Communication styles however, play an essential role in counseling lifestyle [49].

This study explained almost 50% of the variance in lifestyle counseling, which means that the other half is yet to be explained. Many other factors could influence lifestyle counseling behavior. These could be, for example, patient factors, conflict of interest, the role of colleagues and communication styles. Therefore, future research should focus on these other factors that may contribute to its explanation.

First, future research could investigate how the 5A's can be best applied in this context, ideally in terms of effectiveness. This could be done by, for example, performing experimental and/or prospective observational studies among patients. When examining the effectiveness of the 5A's, the role of different approaches and communication techniques on patients' health, motivation and/or intentions can be examined. It is important to take patient characteristics, diseases, lifestyles and into account [50]. For example, it is likely that GPs' lifestyle counseling practices are influenced by the type of chronic disease patients suffer from.

Second, the same research design can be used for examining a possible causal relationship between the determinants and counseling lifestyle. This could be done, for example, by researching one group who were given a lifestyle counseling course and one group without intervention. By performing this research, the influence of GPs' self-efficacy, attitude and subjective norms on their lifestyle counseling practices can be assessed.

Other suggestions for future research include investigating which supporting tools GPs need specifically. It is likely they need different tools for different lifestyle habits and/or diseases; using video-recordings of consultations to collect objective data about GPs' counseling practices; and to study patient experiences regarding lifestyle counseling.

## Supporting information

**S1 File. Survey questionnaire (original version, in Dutch).**
(PDF)

**S2 File. Survey questionnaire (translated into English).**
(PDF)

**S3 File. Data set.**
(SAV)

## Author Contributions

**Conceptualization:** Lisanne Kiestra, Bob C. Mulder.

**Data curation:** Lisanne Kiestra, Bob C. Mulder.

**Formal analysis:** Lisanne Kiestra, Bob C. Mulder.

**Investigation:** Bob C. Mulder.

**Methodology:** Lisanne Kiestra, Bob C. Mulder.

**Project administration:** Lisanne Kiestra.

**Resources:** Iris A. C. de Vries.

**Supervision:** Iris A. C. de Vries, Bob C. Mulder.

**Validation:** Iris A. C. de Vries.

**Writing – original draft:** Lisanne Kiestra.

**Writing – review & editing:** Iris A. C. de Vries, Bob C. Mulder.

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
