## [Decision Letter · Decision Letter 0]

26 May 2020

PONE-D-20-11604

Determinants of lifestyle counselling and current practices: A cross-sectional study among Dutch general practitioners

PLOS ONE

Dear Dr. Mulder,

Thank you for submitting your manuscript to PLOS ONE. After careful consideration, we feel that it has merit but does not fully meet PLOS ONE’s publication criteria as it currently stands. Therefore, we invite you to submit a revised version of the manuscript that addresses the points raised during the review process.

We look forward to receiving your revised manuscript.

Kind regards,

Amir H. Pakpour, Ph.D.

Academic Editor

PLOS ONE

5. We note you have included a table to which you do not refer in the text of your manuscript. Please ensure that you refer to Table 3, 5 and 6 in your text; if accepted, production will need this reference to link the reader to the Table.

6. Please include a copy of Table x which you refer to in your text on page 9.

Reviewers' comments:

Reviewer's Responses to Questions

**Comments to the Author**

1. Is the manuscript technically sound, and do the data support the conclusions?

Reviewer #1: Yes

2. Has the statistical analysis been performed appropriately and rigorously? 

Reviewer #1: Yes

3. Have the authors made all data underlying the findings in their manuscript fully available?

Reviewer #1: Yes

4. Is the manuscript presented in an intelligible fashion and written in standard English?

Reviewer #1: Yes

5. Review Comments to the Author

Reviewer #1: I read the manuscript entitled "Determinants of lifestyle counselling and current practices: A cross-sectional study

among Dutch general practitioners" with great joys. However, I would like the authors to improve their work using the following comments.

1. In Abstract, please mention general practitioners before using the abbreviation of GP.

2. Also in Abstract, Better to mention what the 5 As are after indicating the 5As model. That is, write something like 5As model (Advising, Agree,...).

3. In the Introduction, please use the following reference to talk about the self-behaviors among people with chronic disease.

http://www.shbonweb.com/temp/SocHealthBehav2389-3066382_083103.pdf

4. Please also use the following references to discuss the potential difficulties of people with disability to seek consultation from healthcare providers and the difficulties that healthcare providers may encounter when providing consultation in the Introduction.

http://www.shbonweb.com/temp/SocHealthBehav1254-311131_083833.pdf

Lin, C.-Y., Fung, X. C. C.#, Nikoobakht, M., Burri, A., & Pakpour, A. H. (2017). Using theory of planned behavior incorporated with perceived barriers to explore sexual counseling services delivered by health professionals in individuals suffering from epilepsy. Epilepsy & Behavior, 74, 124-129.

5. Line 142. Please describe Theory of Planned Behavior with more details. Specifically, the authors should mention that the Theory of Planned Behavior has been widely used. Please refer to the following references.

Lin, C.-Y., Oveisi, S., Burri, A., & Pakpour, A. H. (2017). Theory of Planned Behavior including self-stigma and perceived barriers explain help-seeking behavior for sexual problems in Iranian women suffering from epilepsy. Epilepsy & Behavior, 68, 123-128.

Lin, C.-Y., Updegraff, J. A., & Pakpour, A. H. (2016). The relationship between the theory of planned behavior and medication adherence in patients with epilepsy. Epilepsy & Behavior, 61, 231-236

Hou, W.-L., Lin, C.-Y., Wang, Y.-M., Tseng, Y.-H., & Shu, B.-C. (2020). Assessing Related Factors of Intention to Perpetrate Dating Violence among University Students Using the Theory of Planned Behavior. International Journal of Environmental Research and Public Health, 17(3), 923.

Lin, C.-Y., Cheung, M. K. T., Hung, A. T. F., Poon, P. K. K., Chan, S. C. C., & Chan, C. C. H. (2020). Can a Modified Theory of Planned Behavior Explain the Effects of Empowerment Education for People with Type 2 Diabetes? Therapeutic Advances in Endocrinology and Metabolism, 11, 1-12.

Fung, X. C. C., Pakpour, A. H., Wu, K.-Y., Fan, C.-W., Lin, C.-Y., Tsang, H. W. H. (2019). Psychosocial variables related to weight-related self-stigma in physical activity among young adults across weight status. International Journal of Environmental Research and Public Health, 17, 64.

Cheng, O. Y., Yam, C. L. Y., Cheung, N. S., Lee, P. L. P., Ngai, M. C., & Lin, C.-Y. (2019). Extended Theory of Planned Behavior on eating and physical activity. American Journal of Health Behavior, 43(3), 569-581.

6. The authors mentioned that they calculated the Conbach's alpha. However, I did not see the results. Maybe I overlooked?

7. It is weird that the Table begins from 3, then goes to 5 and 6.

8. Please also provide the unstandardized coefficient and SE for the regression models in Table 5.

6. PLOS authors have the option to publish the peer review history of their article (what does this mean?). If published, this will include your full peer review and any attached files.

Reviewer #1: No

---

## [Author Response · Author response to Decision Letter 0]

16 Jun 2020

Dear Dr Pakpour and Reviewer,

We would like to thank you for the opportunity to revise and resubmit our paper ‘Determinants of lifestyle counseling and current practices: A cross-sectional study among Dutch general practitioners’. We are very grateful for you careful reading of our manuscript, and the resulting comments that both of you provided, as these have allowed us to improve our manuscript. 

We have appended all comments below, and have responded to each comment by explaining the changes we have made in the manuscript. 

Thank you again for considering our manuscript for publication in PLOS ONE.

Sincerely,

Bob Mulder (corresponding author), Lisanne Kiestra and Iris de Vries

 

Journal requirements brought forward by the academic editor, Dr. Pakpour

Response: Thank you for your suggestion. As with the original manuscript, we have carefully checked and applied PLOS ONE’s manuscript requirements.

Response: We agree that more information can be supplied to allow replication of our study. Therefore, we have added the survey questionnaire that we used in our study as Supporting Information, both in the original language (i.e. Dutch) and English. We cite the supporting information files on p. 5, line 114.

Response: We fully support PLONE ONE’s policy on sharing data, and after careful consideration we have concluded there are no restrictions on sharing our dataset. Please see our response to the next comment (‘b’) for further explanation.

Response: As noted, there are no restrictions; therefore, we have indeed decided to upload the minimal anonymized data set that is necessary to replicate study findings, as Supporting Information file, S3 File. This supporting information file is not cited in text. 

Response: Thank you for offering multiple options to improve the presentation of this preparatory analysis in the manuscript. As noted in our response to comment #3, we have decided to share the dataset as Supporting Information file. As a result, we were able to delete the phrase “data not shown” from our manuscript. After careful scrutiny, we decided to slightly alter the wording of this section, in order to improve clarity. 

Old text, page 12, lines 262-266: 

“A separate regression analysis (data not shown) showed that being a member of ‘Vereniging Arts en Leefstijl’ and gender were the only background variables significantly associated with lifestyle counseling. Therefore, these variables were maintained as covariates throughout the regression analysis. Age and working days were not significantly associated with lifestyle counseling in any of the models and were therefore excluded.”

New text, page 13, lines 273-278:

“The regression analysis that aimed to test associations with background variables showed that only being a member of ‘Vereniging Arts en Leefstijl’ and gender were significantly associated with lifestyle counseling. Therefore, these variables were maintained as covariates throughout the main regression analysis. Age and working days were not significantly associated with lifestyle counseling in any of the models and were therefore excluded.”

5. We note you have included a table to which you do not refer in the text of your manuscript. Please ensure that you refer to Table 3, 5 and 6 in your text; if accepted, production will need this reference to link the reader to the Table.

Response: Unfortunately, the numbering of the tables, as well as in-text references to the tables, were below par. We apologize for these errors and any misunderstanding they have caused. All in all, there are three tables in the manuscript; of course, these need to be numbered Table 1, 2 and 3. In the revised manuscript, this is now the case. Please also note that we have added two table references:

- Page 12, line 262/263: “(see Table 2 for a complete overview of correlations).”

- Page 14, line 287: “(see Table 3).”

6. Please include a copy of Table x which you refer to in your text on page 9.

Response: As noted in our previous response, the original manuscript contained errors in the numbering of – and referring to – tables. “Table x” should have been “Table 1”, and this has been corrected in the revision (see page 10, line 214 of the revised manuscript).

Reviewers' comments

Reviewer #1

I read the manuscript entitled "Determinants of lifestyle counseling and current practices: A cross-sectional study among Dutch general practitioners" with great joys. However, I would like the authors to improve their work using the following comments.

1. In Abstract, please mention general practitioners before using the abbreviation of GP.

Response: First, we would like to thank the Reviewer for the compliment on our manuscript. We are also grateful for the Reviewer’s efforts to carefully read and review our manuscript. We believe that the manuscript has greatly benefitted from the Reviewer’s work.

As requested, we have now written out “general practitioners” followed by “(GPs)” as abbreviation. Please see the Abstract on page 2, line 25/26.

2. Also in Abstract, Better to mention what the 5 As are after indicating the 5As model. That is, write something like 5As model (Advising, Agree,...).

Response: We agree that it is better to immediately list all the As, and have changed the text accordingly. Please see the Abstract on page 2, line 28.

3. In the Introduction, please use the following reference to talk about the self-behaviors among people with chronic disease.

http://www.shbonweb.com/temp/SocHealthBehav2389-3066382_083103.pdf

Response: Thank you for suggesting this paper. Unfortunately, the hyperlink as given by the reviewer leads to a ‘page not found’ announcement on the website of the journal ‘Social Health and Behavior’. After searching this journal’s archive, we have found an interesting editorial that we believe is the one the reviewer wanted to point out: Pakpour AH, Lin CY, Alimoradi Z. Social health and behavior needs more opportunity to be discussed. Soc Health Behav2018;1:1. We have referred to this paper in the Introduction (page 3, line 53).

4. Please also use the following references to discuss the potential difficulties of people with disability to seek consultation from healthcare providers and the difficulties that healthcare providers may encounter when providing consultation in the Introduction.

http://www.shbonweb.com/temp/SocHealthBehav1254-311131_083833.pdf

Lin, C.-Y., Fung, X. C. C.#, Nikoobakht, M., Burri, A., & Pakpour, A. H. (2017). Using theory of planned behavior incorporated with perceived barriers to explore sexual counseling services delivered by health professionals in individuals suffering from epilepsy. Epilepsy & Behavior, 74, 124-129.

Response: Thank you for pointing out this relevant paper, which we now cite in the Introduction (page 3, line 68), and also in the Methods section (p. 7, line 150)

5. Line 142. Please describe Theory of Planned Behavior with more details. Specifically, the authors should mention that the Theory of Planned Behavior has been widely used. Please refer to the following references.

Lin, C.-Y., Oveisi, S., Burri, A., & Pakpour, A. H. (2017). Theory of Planned Behavior including self-stigma and perceived barriers explain help-seeking behavior for sexual problems in Iranian women suffering from epilepsy. Epilepsy & Behavior, 68, 123-128.

Lin, C.-Y., Updegraff, J. A., & Pakpour, A. H. (2016). The relationship between the theory of planned behavior and medication adherence in patients with epilepsy. Epilepsy & Behavior, 61, 231-236

Hou, W.-L., Lin, C.-Y., Wang, Y.-M., Tseng, Y.-H., & Shu, B.-C. (2020). Assessing Related Factors of Intention to Perpetrate Dating Violence among University Students Using the Theory of Planned Behavior. International Journal of Environmental Research and Public Health, 17(3), 923.

Lin, C.-Y., Cheung, M. K. T., Hung, A. T. F., Poon, P. K. K., Chan, S. C. C., & Chan, C. C. H. (2020). Can a Modified Theory of Planned Behavior Explain the Effects of Empowerment Education for People with Type 2 Diabetes? Therapeutic Advances in Endocrinology and Metabolism, 11, 1-12.

Fung, X. C. C., Pakpour, A. H., Wu, K.-Y., Fan, C.-W., Lin, C.-Y., Tsang, H. W. H. (2019). Psychosocial variables related to weight-related self-stigma in physical activity among young adults across weight status. International Journal of Environmental Research and Public Health, 17, 64.

Cheng, O. Y., Yam, C. L. Y., Cheung, N. S., Lee, P. L. P., Ngai, M. C., & Lin, C.-Y. (2019). Extended Theory of Planned Behavior on eating and physical activity. American Journal of Health Behavior, 43(3), 569-581.

Response: We agree that these papers present relevant examples of the wide use and predictive properties of the theory of planned behavior. However, not all papers concern health behaviors of specific patient populations. We do feel it is important to focus our arguments and related references on that domain. This means we have cited all suggested articles but for the articles by Fung et al. (2019) and Hou et al. (2020).

In addition, we have revised the description of “Determinants of counseling practices”, under “Measures” in the Methods section. This was done to better explain the theory of planned behavior, as well as to incorporate the new references.

Old text, page 7, lines 142-145:

“The determinants of lifestyle counseling practices, as well as their measurement, were derived from prior studies in this domain [11,12,22], and the theory of planned behavior [23]. The main determinants were attitude, subjective norms, and perceived behavioral control. For each determinant, (...)”

New text, page 7, lines 146-152:

“The determinants of lifestyle counseling practices, as well as their measurement, were derived from prior studies in this domain [13,14,22], and the theory of planned behavior [25]. The theory of planned behavior has been widely used to predict health behaviors of diverse patient populations [26,27,28,29], and – to a much lesser extent – to predict provider counseling behaviors [12]. According to the theory of planned behavior, the main determinants of behavioral intention are attitude, subjective norms, and perceived behavioral control. For each determinant, (...)”

6. The authors mentioned that they calculated the Conbach's alpha. However, I did not see the results. Maybe I overlooked?

Response: No, the reviewer did not overlook, because Cronbach’s alphas were indeed not reported in the original manuscript. Of course, it is good scientific practice to report all alphas where relevant. This is why we revised the manuscript and added Cronbach’s alpha for all multiple-item scales. Specifically, we now report Cronbach’s alpha in the Methods section, subsection “Measures”, for:

- The full 5As scale, on p. 6, line 129;

- Attitude, on p. 7, lines 154;

- Self-efficacy, on p. 7, line 164.

Please, also note that we slightly rephrased the description of how we measured the 5As on p. 6, notably lines 125 – 128; line 132; lines 135 – 136, and lines 138 – 139.

7. It is weird that the Table begins from 3, then goes to 5 and 6.

Response: We agree; unfortunately the numbering of tables contained errors. Please see our responses to the editor’s comments #5 and #6 for the revisions we made regarding the numbering of – as well as referring to – tables. 

8. Please also provide the unstandardized coefficient and SE for the regression models in Table 5.

Response: We have extended the table that presents the results of the regression model (called Table 3 in the revised manuscript) with the data as requested by the reviewer.

---

## [Editor Report · Decision Letter 1]

26 Jun 2020

Determinants of lifestyle counseling and current practices: A cross-sectional study among Dutch general practitioners

PONE-D-20-11604R1

Dear Dr. Mulder,

We’re pleased to inform you that your manuscript has been judged scientifically suitable for publication and will be formally accepted for publication once it meets all outstanding technical requirements.

Kind regards,

Amir H. Pakpour, Ph.D.

Academic Editor

PLOS ONE
---

## [Editor Report · Acceptance letter]

8 Jul 2020

PONE-D-20-11604R1 

Determinants of lifestyle counseling and current practices: A cross-sectional study among Dutch general practitioners 

Dear Dr. Mulder:

I'm pleased to inform you that your manuscript has been deemed suitable for publication in PLOS ONE. Congratulations! Your manuscript is now with our production department. 

Kind regards, 

on behalf of

Dr. Amir H. Pakpour 

Academic Editor

PLOS ONE